# Are Peach Cultivars Used in Conventional Long Food Supply Chains Suitable for the High-Quality Short Markets?

**DOI:** 10.3390/foods10061253

**Published:** 2021-05-31

**Authors:** Cosimo Taiti, Corrado Costa, William Antonio Petrucci, Laura Luzzietti, Edgardo Giordani, Stefano Mancuso, Valter Nencetti

**Affiliations:** 1Department of Agriculture, Food, Environment and Forestry (DAGRI), University of Florence, Viale delle Idee 30, Sesto F.no, 50019 Florence, Italy; antoniopetrucci741@gmail.com (W.A.P.); edgardo.giordani@unifi.it (E.G.); stefano.mancuso@unifi.it (S.M.); valter.nencetti@unifi.it (V.N.); 2Consiglio per la Ricerca in Agricoltura e L’analisi Dell’economia Agraria (CREA)—Centro di Ricerca Ingegneria e Trasformazioni Agroalimentari, Monterotondo, 00189 Rome, Italy; corrado.costa@crea.gov.it; 3Centre for Organismal Studies (COS) Heidelberg, Heidelberg University, 69120 Heidelberg, Germany; laura.luzzietti@cos.uni-heidelberg.de

**Keywords:** *Prunus persica*, food supply chains, cluster analysis, VOCs

## Abstract

Peaches are climacteric and highly perishable fruits, with a short shelf life, and are prone to rapid deterioration after harvest. In this study, the chemical proprieties, aroma profile and a sensory evaluation were conducted to: (1) characterize and compare fruits of 13 different peach and nectarine cultivars, harvested at physiological maturation; and (2) assess the suitability of these cultivars, that are successfully used in long food supply chains (LFSCs), for their use in short food supply chains (SFSCs). Through statistical analysis clear differences were found among the studied cultivars, and in particular between cultivars suited to SFSCs compared to those suited for LFSCs. Results indicate that, despite all cultivars being planted in the same orchards and with the same pre-harvest management and practices, their post-harvest performances were mainly influenced by the cultivar genetic makeup. Therefore, cultivars conventionally used in SFSCs, such as “Guglielmina” and “Regina di Londa”, had the best aroma, sweetness and juiciness compared to LSCPs ones. In contrast, the LSCPs varieties showed interesting values for firmness and crunchiness.

## 1. Introduction

In the recent years, the fresh fruit market has grown significantly fueled by an increasing demand of customers concerned in healthier diets, food quality requirements and year around availability [1]. In particular, the increased consumption of fresh products, fruits included, is mainly motivated by consumer changes in the lifestyle together with the increased awareness on health promoting properties associated with fresh fruit consumption [2]. Therefore, the practice of selling through Short Food Supply Chains (SFSCs) has been of interest and has undergone significant developments in the last decade, spurring the interest of producers and consumers whit economic, social and environmental benefits across Europe and North America. Indeed, fruit and vegetable short supply chains, which are derived from US and Anglo-Saxon experiences such as “pick-your-own” and “farmer’s market” or from Northern European countries such as the “box schemes”, have emerged in all of Italy [3]. Actually, farmers obtain some added value from their products through the SFSCs, while consumers get better foods “with the farmers’ face on it” [4] and, on the other hand, SFSCs promotes the rise of a new more territorially based rural development [5], the adoption of more environmentally friendly rural agronomic practices and a more suitable use of local resources [6,7]. In this context SFSCs represent different ways of producing, distributing, retailing and buying foods [8] and therefore, they are commonly referred as alternative food quality markets [9]. Moreover, as reported elsewhere [10,11] French and Italian, and, in general, EU farmers and consumers are increasingly turning to SFSCs in order to get the positive effects related to the quality of food products, the improvement of the environment and the development of local economy [12,13].

Peaches (*Prunus persica* L. Batsch) fruits are highly perishable, showing a short post-harvest stage and very limited shelf life. Hence, peaches, similarly to most stone fruits, are prone to rapid deterioration after harvest and not suitable, in contrast to other fresh pulp fruits (e.g., apple and pear), for long post-harvest storage, thus limiting long distance export. Peach breeders and producers have tried to release and adopt, respectively, cultivars producing fruits with long post-harvest life, high firmness and good appearance (shape, color and texture) to satisfy especially the requirements of the large-scale retail channel [14,15]. On the contrary, organoleptic (as flavor and aroma) and nutritional properties were often considered as secondary aims and have only been slightly improved or even lost over time [16]. Moreover, since peach quality specifications for large-scale retail channels mainly rely on fruit size and peel color, fruits are harvested at commercial maturity stage in order to increase storage time, market life and to minimize fruit physical damages [17], thus avoiding the need to manage peaches collected at the optimal maturity stage related to organoleptic quality (i.e., at the physiological ripening) that would be too soft to be handled and will have a very short storage and shelf life. As a consequence, important quality attributes linked to the maturation process affecting consumer’s requirements and satisfaction (i.e., sugar content, aroma and flavor), cannot be satisfied by the large-scale retail chain [18,19]. Accordingly, as reported by Della Casa [20], the most frequent consumer complaints about the quality of peaches and nectarines regard the lack of flavor and excessive firmness, which are associated to premature fruit harvest [21]. Contrary to the Long Food Supply Chains (LFSCs), in the Short Food Supply Chains (SFSCs) fruits are selected and harvested when they have reached a more advanced maturation on trees, with a relevant increase in fruit quality. The main objective of the present study was to assess the suitability for short post-harvest management of a set of international peach and nectarine cultivars adopted successfully in long post-harvest supply chains.

## 2. Results and Discussion

### 2.1. Morphological Fruit Traits

Morphological information about the 13 varieties used in this study are reported in Table 1. As expected, great variations were observed in the morphological traits (namely shape, size and flesh color) among the studied varieties. Fruit shape varied from “Round” (ten cultivars) to “Ovate” (three cultivars), while fruit weight showed a wide range, with the lowest values represented by “Nectaross” (average weight 170 ± 10 g) and the highest by “Lady Erica” (312 ± 15 g) and “Alma” (295 ± 25 g). “Nectaross” fruits showed the lowest value for all size parameters (diameter and height), representing by far the smallest sample included in this study. Fruit ground skin color ranged from “Greenish-white” to “Not visible”, while overskin color was red. Between peach and nectarines grand total no significant differences were highlighted in morphological traits.

### 2.2. Analysis of Physico-Chemical Measures

Quality is composed of physicochemical characteristics that give a product consumer appeal and acceptability [22]. Therefore, the key factors that determine high-quality in peaches and nectarines are the following: skin appearance, firmness, flavor and aroma, sugar and acid content and their ratio [23]. The physicochemical attributes of the studied cultivars are reported in Table 2.

Firmness at harvest differed considerably among cultivars and in agreement with other authors [24,25]; peach cultivars have shown a lower average flesh firmness than nectarines (2.18 and 3.72 kg, respectively) while the softer ones were “Regina di Londa” and “Maria Marta” (1.53 ± 0.5 and 1.60 ± 0.5 kg, respectively). More in detail, cultivars with the highest value of fruit hardness were the two nectarines “Big Top” (4.30 ± 0.5) and “Alma” (4.20 ± 0.4). As reported by Dirlewanger et al. [26], the variation in fruit quality at harvest is related to many interrelated factors. However, sugar and acid content and their ratio, are important factors to define peach and nectarine quality [23]. pH ranged between 3.1 (±0.4) observed in “Big Top” to 5.20 (±0.3) in “Sweet Dream”. At the harvesting time all the cultivars analyzed had reached an SSC higher than the value (10% °Brix) proposed by Kader [27] as the minimum quality standard. The lower value of SSC was observed in “Sweet Dream” (11.7 ± 0.6 °Brix), while the highest was observed in “Regina di Londa” (16.4 ± 1.4 °Brix) and “Nectaross” (16.0 ± 1.2 °Brix). Moreover, TA content was very variable among the analyzed cultivars (Table 2), the lowest values were observed in “Sweet Dream” (8.6 ± 0.6), “Lady Erica” (11.4 ± 0.3) and “Alma2” (13.2 ± 1.1), while the highest values were found in “Nectaross” (47.8 ± 2.7), “Alma” (42.3 ± 0.3) and “Venus” (35.5 ± 1.1). Moreover, the highest values of SSC/TA were observed in the fruit juice of “Sweet Dream” and “Lady Erica” (1.36 and 1.24, respectively), while the lowest were found in “Alma”, “Venus” and “Nectaross” (0.31, 0.33 and 0.34, respectively). 

The PCA has been performed on the data set derived from the physicochemical measures (Table 2) and was represented in Figure 1. The first two components obtained with PCA explained more than 51% of the total variability and the derived two-dimensional plot provided the separation of the thirteen peach samples. Therefore, the PCA give a general overview of each variety’s ordination and show how the SFSCs varieties are positioned on the negative portion of the second axis in relation to high sugar amounts and low penetrometer values, since they are characterized by a more tender pulp than those cultivars adopted in LFSCs with a quite hard flesh. 

In fact, “Regina di Londa” and “Guglielmina” are two traditional varieties not used in the large-scale distribution. Furthermore, some differences were observed also between the two SFSC peaches. Particularly, “Regina di Londa” (a variety belonging to the “Burrone Fiorentine group”) showed a very soft pulp and a very high TA content compared to “Guglielmina” (a selection from the “Cotogne Fiorentine”) [28]. As reported by Bellini [28], “Burrone Fiorentine” and “Cotogne Fiorentine” are two old late ripening peach types used exclusively for fresh consumption. The fruits of “Burrone Fiorentine” are white, firm, very sweet, pleasant, aromatic and fragrant, whilst, in “Cotogne Fiorentine” the pulp is yellowish orange, considerable consistency and sometimes crunchy, with a fairly sweet flavor and muscat aroma.

### 2.3. Rapid Detection of Volatile Compounds Using PTR-ToF-MS

The VOCs (Volatile Organic Compounds) emitted by fruits of twelve different variety of peach were acquired by the PTR-ToF-MS tool. Each mass spectra were generated by the mean of twelve fruit replicates of each peach varieties analyzed. Then, the mass spectral data were used as fingerprints. Therefore, the peaks detected in each sample and their corresponding signal intensities (expressed as ppbv) have been employed as a pattern for sample comparison. By the peak’s extraction were detected a total of 66 tentatively identified compounds in the range of measured masses (ranged from *m*/*z* 25 and *m*/*z* 230), which derived from the protonation of various VOCs (Table A1). Among them, a minimum of 45 peaks were detected in “Alma2” and “Nectaross”, while a maximum of 57 peaks in “Maria Marta” (Table A1). Among these peak signals, stand out the peaks linked to the ripening process as *m*/*z* 33.033 (TI: methanol), 45.033 (TI: acetaldehyde), 47.049 (TI: ethanol) and 61.028 (TI: acetic acid), respectively [21]. The methanol and acetaldehyde emission were high and comparable, in all peach samples and proving probably how the maturity stage of all varieties was similar (data not shown). Moreover, as a general overview, among the cultivars there are difference in many VOC types and intensities (Figure 2, Table A1). As in other fruits, VOC emission by peach is influenced by many factors, such as: cultivar, tissue, processing, storage, ripening stage, harvest and environmental conditions [29]. Moreover, as reported elsewhere [30,31] peach aroma and flavor are linked to the biosynthesis of different phytochemical compounds and thus their biosynthesis is reliant on primary and secondary metabolites derived by the photosynthetic process [32]. Indeed, carbohydrates, fatty acids and amino acids are the direct precursors of many volatiles that significantly contribute to the fruit aroma [17]. Saturated and unsaturated fatty acids generate most plant volatiles, such as esters, ketones, lactones, aldehydes and alcohols [33]. For example, from methionine or amino acids degradation yields aldehydes and alcohols, which confer green and unripe aromatic notes to the fruits [31]. Therefore, in agreement with Carrari and Fernie [32], the differences in levels and types of aromatic compounds detected among the peach variety studied may have been affected by the number of phytochemical compounds. Aroma is a complex mixture of many volatile compounds, and since the VOCs composition is specific to species and often to the variety of fruit [17] by our analysis it emerged as the varieties under study showed a rather specific volatile profile (Figure 2, Table A1). Aubert and Milhet [29], reported more than 100 volatile compounds in peach, which include C6 aldehydes, mono and sesquiterpenes, alcohols, lactones and esters. Furthermore, it is known as some ester compounds (i.e., hexyl acetate and (Z)-3-hexenyl acetate) are considered the odorants most influencing the aroma and flavor quality of peach fruit [34], together with c- and d-decalactone, C6 compounds, alcohols, terpenoids and phenolic volatiles [35]. Generally, C6 aldehyde compounds and alcohols offer the green-note aroma, while lactones and esters are responsible for fruity aromas, finally the esters that are the predominant class of volatile compounds in many ripe fruits providing a fruity note [31]. Figure 2 reports the intensity of the principal odorant VOCs (C6, ester, lactone and terpene compounds) detected for each variety included in our study. It is interesting to note that, many aroma compounds, in particular lactones and terpenes, were absent particularly in “Nectaross” and “Alma2”, while “Big Top” showed absence or very low presence of ester compounds. On the contrary, other varieties suitable for LFSCs such as “Venus”, “Rome Star” and “Romagna Big” showed the highest intensity of monoterpene compounds (*m*/*z* 137.132) and high emission of lactones (*m*/*z* 115.075). On the other hand, the peach suitable for SFSCs (“Regina di Londa” and “Guglielmina”) showed the highest intensity of terpenoid compounds (*m*/*z* 153.127) and high intensity of some ester (e.g., *m*/*z* 89.059) and lactone compounds (*m*/*z* 115.075). Thus, the presence or the high intensity of some compounds (i.e., *m*/*z* 89.059, 115.075, 117.091, 153.127 and 159.140) detected in “Regina di Londa” and “Guglielmina” are probably the reason for their pleasant, aromatic and fruity aromas (Figure 2).

### 2.4. Sensory Measurements

#### 2.4.1. Panel Test

Fruit quality is composed by a combination of sensory, nutritional, chemical, mechanical and functional properties [36]. Fruit flavor is a complex combination among sweetness, sourness and aroma [30,31], and these quality attributes mainly depend on the content of sugars, organic acids and aroma volatiles [37]. Colaric et al. [38] in a previous article on peach and nectarine quality attributes, showed that the sensory evaluation is an efficient method to define fruit quality and to discriminate among different cultivars. The panel test results are showed in Figure 3, where the average scores of the ten attributes evaluated by panelists were reported. 

Therefore, among the estimated peach samples, both the varieties suitable for the SFSCs were preferred by panelists and rated as the best. Indeed, the yellow-fleshed peach “Guglielmina” and the white-fleshed peach “Regina di Londa” were evaluated the best as highly aromatic, very tasteful, very juicy and sweet. Conversely, all the LSCPs were evaluated of lower quality. Among these, yellow-fleshed peach “Nectaross” showed the highest score for overall rating, aroma intensity, acidity and juiciness. Instead, “Sweet Dream” and “Alma 2” received the lowest values for overall rating and aroma but at the same time were the hardest and the crispiest varieties. Since all the peaches included in this study were collected at physiological maturation, the high values observed in SSCPs for overall judgment, aroma, sweetness and juiciness could be linked to genetic factors, since all the trees were cultivated in the same orchard, while the LSCPs are known to have an acceptable taste when collected at the usual commercial maturation stage, but as shown in this study they do not reach the SSCPs in fruit quality. Indeed, peach breeding programs of the past 30–40 years have been focused on traits associated with fruit appearance, hardness (i.e., fruit size, color and firmness) and shelf-life, leaving out the quality attributes [39]. Furthermore, it is worth noticing that breeders did not select the progenies on the basis of fruit quality at physiological maturation because the short supply chain market was not taken into account. Indeed, nowadays, the Large Retail Organization imposes a long shelf life for the fruits and thus the old varieties have disappeared from the market. Moreover, in order to guarantee a long shelf-life the fruit harvest must be carried out at a high level of hardness of the pulp, anticipating the physiological maturation. In particular, for the Large Retail Organization the climacteric fruit are harvested when commercial maturity is reached, since the ripening process continues during post-harvest shipping and marketing. This situation is linked to the past customs of consumers when the fruits was harvested only at physiological maturation because they were consumed one or two days after the harvest and therefore it was not necessary to preserve them for a long time. Therefore, an increasing trend to reintroduce local varieties into SSCPs has been recently observed [40]. The different quality of peach studied has been also shown by applying the PCA analysis (Figure 4) where the SSCPs are positioned on the positive portion of the first axis in relation to high values of overall judgment, aroma, sweetness and juiciness. As reported by Mennone et al. [41], “Regina di Londa” and “Guglielmina” are characterized by (1) sublime tastes and special aromas only when reached the physiological maturation and (2) a very short shelf-life.

#### 2.4.2. Consumer Acceptability

The consumer acceptance was calculated on 13 different peach and nectarine varieties as the percentage of respondents who liked the sample, with scores >5, and the result are reported in Table 3. 

Each consumer was asked to use all their senses (sight, smell, taste, touch and even hearing) to evaluate quality of fruits. Therefore, the consumer integrates all those sensory inputs such as appearance, aroma, flavor, hand-feel, mouthfeel and chewing sounds, into a final judgment of the acceptability of that fruit or vegetable [42]. For all varieties analyzed, the degree of liking and percentage of consumers that accepted the fruit has always been more than 50%.

Consumer acceptance, expressed as the degree of liking and the percentage of satisfied consumers, was higher for the varieties suitable to SFSCs compared to the LSCPs. In particular, “Regina di Londa” and “Guglielmina” showed both a higher degree of liking (8.36 and 8.73, respectively) and that they got the full acceptance by all the consumers included in this study (Table 3). Moreover, among the modern cultivars, “Nectaross” (86% of the interviewees) and “Lady Erica” (78% of the interviewees) have shown a higher acceptation level, while “Nectaross” (7.43) and “Maria Marta” (7.00) showed a higher degree of liking. On the contrary, “Alma” (6.40), “Alma2” (6.12) and “Sweet Dream” (6.08) have showed a low degree of liking and also were evaluated the worst by the consumers (Table 3).

### 2.5. Correlation between Aroma Descriptors, Physico-Chemical Parameters and Volatile Compounds

The Canonical Correspondence Analysis (CANOCO), conducted on physical–chemical and Panel test matrices (Figure 5), evidenced a position of the SSCPs on the positive side of the second axis. From the panel test variables point of view (green vectors), overall judgment, aroma, sweetness and juiciness returned a positive contribute to SSCPs, while firmness and crunchiness returned a negative one. Since the SSCPs tend to have a short shelf life compared to the LSCPs, the negative contribution of firmness and crunchiness on the SSCPs was awaited. Indeed, as reported in Table 2 and Figure 3, SSCPs highlighted low values both of firmness and crunchiness; on the contrary, by pooling the entire dataset obtained, we can see how the SSCPs varieties showed a much higher quality level (either pomological, aromatic and sensorial) than LSCPs (Figure 2, Table 2 and Appendix A Table A1).

Subsequently, the CANOCO applied on VOCs and Panel test matrices (Figure 6), evidenced a position of the SSCPs on the positive side of the first axis. This positioning and the positive contribute of the panel test variables (overall judgment, aroma, sweetness and juiciness) are like the ones observed by the PCA. Therefore, by pooling the VOCs and the panel test results, it was possible to identify those few aromatic molecules able to have influenced the panel evaluation. High values of signals detected at *m*/*z* 43.018 (TI: Fragment (ester), 61.028 (TI: acetic acid), 89.059 (TI: ethyl acetate), 145.122 (TI: ethyl hexanoate/hexyl acetate), 153.127 (TI: terpenoid like-compounds) and 177.185 (TI: ethyl acetate cluster) better characterized the SSCPs.

In general, the signals detected at *m*/*z* 43.018 (TI: ester fragment), 89.059 (TI: ethyl acetate), 145.122 (TI: ethyl hexanoate/hexyl acetate) and 177.185 (TI: ethyl acetate cluster), all compounds belonging to the ester compound, are the highest intensity in SSCPs compared to the LSCPs (Figure 2). The highest intensity of these esters’ compounds, contributes significantly to give fruity and pleasant aromas and flavors that are typical of SSCPs. In particular, ethyl acetate (*m*/*z* 89.059) has a fruity smell with a brandy note and is the most common ester in fruits; whilst the ethyl hexanoate and/or hexyl acetate (*m*/*z* 145.122) have, respectively, a fruity aroma similar to sweet pineapple, green banana or fruity green sweet apple (www.thegoodscentscompany.com, accessed on 13 October 2020). Finally, the signal detected at *m*/*z* 153.127 (TI: terpenoid-like compound) identifies aromatic compounds and in particular a mixture of terpenoids or other aromatic compounds with molecular formula C_10_H_16_O (Table A1).

## 3. Methodology

### 3.1. Fruit Collection

In this study, carried out in 2019, a total of 13 *P. persica* genotypes (2 of them currently adopted for SFSC in Italy, and 11 marketed in the LFSCs) were used. General information of the studied genotypes is reported in Table 4. The set includes Italian and internationally high value peach and nectarine cultivars, representing a wide range of ripening time and the worldwide most appreciated typologies (i.e., yellow and white fleshed peaches and nectarines, with titratable acidity between 70 and 130 meq/L) [43], as well as two important old peach varieties belonging to the germplasm of Tuscany (Italy), namely “Regina di Londa” and “Guglielmina”. “Regina di Londa” is a white peach belonging to the ancient peach types called “Burrone Fiorentine” (a variety spread in Tuscany in the past and characterized by low pulp firmness) while “Guglielmina”, belongs to the yellow flesh “Cotogne Fiorentine” group, characterized by a rather compact pulp [28]. Both “Regina di Londa” and “Guglielmina” fruits are traditionally marketed in the SFSCs and consumed within 3–4 days after harvesting, while the fruits belonged to the 11 cultivars obtained by modern breeding programs, are used essentially in LFSCs, being consumed approximately after 20 days from harvesting.

Trees were cultivated at ≈200 m asl in “Cimatti Enea Farm” (44°15′28.5″ N 11°55′01.4″ E) located in Faenza (Ravenna Province—Emilia Romagna Region). All the trees (6–8 years old plants grafted on GF677) were grown at 5 × 3.5 m distance and trained following the “palmette system” typical of the area. Weed control, irrigation and fertilization were carried out following the guidelines for peach cultivation of Emilia-Romagna Region (https://agricoltura.regione.emilia-romagna.it, accessed on 20 September 2018). The average yields ranged between 20 to 22 t/ha depending on the cultivar. All fruits were harvested from the southern or western part of the crown of three adult trees and within 1.5–2 m height above the ground. Fruits were collected from July to September (Table 4). For each cultivar, based on the farmer’s experience, fruits were harvested when ready for a short distance market (i.e., to be consumed after 3–4 days from collection, corresponding to a pulp firmness lower than 4.5 kg); this corresponds approximately a harvesting postponed of one week respect to the conventional commercial harvesting time for LFSC cultivars. All fruits were selected through a visual inspection conducted to avoid interference of different sizes and presences of defects with the analysis. A total of 80 homogeneous fruits per cultivar were collected: 45 fruits for sensory analysis (sensory evaluation and consumer test), 25 for the physicochemical analyses and 10 for spectrophotometric analyses (PTR–ToF–MS). All the fruits immediately harvested were transported to the laboratory of the Department of Agriculture, Food, Environment and Forestry—University of Florence, stored in a climatic chamber (5 ± 1 °C, 90% relative humidity) for one day and subsequently utilized for physicochemical and sensorial analysis.

### 3.2. Analysis of Physicochemical Measures

*Phenotypic variability.*Table 4 report a general information such as: fruit type, fruit flesh color, fruit shape, diffusion area, ripening time, length of supply chain and perceived acidity of each variety included in this study.

*Firmness.* Firmness (expressed as kgf) was measured as the force needed to penetrate fruit pulp by an 8 mm diameter plunger using a hand penetrometer (Model 53200, Turoni, Italy).

*Total soluble solids (SSC) and Titratable acidity (TA).* The TSS concentration for each sample of peach fruit was determined squeezing 15 g of pulps and putting some drops on an N1 Atago hand refractometer for reading (Atago Co., Tokyo, Japan); results were expressed as °Brix. Titratable acidity (TA) was measured on 15 g of pulp blended with 150 mL of distilled water; the solution was filtered, and the TA was determined by the AOAC method [45] and expressed as meq 100 g of fresh weight. Furthermore, TSS and TA data have been used to assess the SSC/TA ratio.

### 3.3. Rapid Detection of Volatile Compounds Using PTR-ToF-MS

For VOCs analysis PTR-ToF-MS (PTR-MS 8000 model, Ionicon Analytik GmbH, Innsbruck, Austria) were used, in its standard configuration, following method and instrumental setting previously described by Taiti and co-authors in a previous work [46]. The headspace VOCs emitted from each sample were measured by placing a single sample in a 1 L glass jar (Bormioli Srl., Bologna, Italy), which had a glass lid provided with inlet and outlet Teflon pipes, which connect, respectively, the chamber to the PTR-ToF-MS system and to the zero-air generator. For each variety, each sample consisted by a half peach (without core) obtained from twelve different peaches [n = 20 replicates (10 × 2) × 13 varieties]. In particular, each fruit was cut in two pieces, removed from the core, and each half was inserted in a different glass jar for the VOCs analysis. The values obtained from the two halves of each fruit have been averaged before data analysis.

For all samples, prior to measurement the glass jar containing the sample was incubated for 60 s. Moreover, the blank measurements were carried out after analyzing three fruit samples, while between each sample 1 min interval was kept to avoid memory effects. The range of mass spectra were recorded at 20–230 *m*/*z*, and throughout the experiment the operating parameters were maintained constant. In particular, the standard operating conditions were: 0.1 ns per channel sampling time, pressure 2.3 mbar, temperature 60 °C and voltage 600 V, extraction voltage at the end of the pipe (Udx) 35 V and corresponding to an E/N value of 137 Td (Td: Townsend; 10^−17^/V cm^2^ s), which gives a good balance between excessive water cluster formation and product ion fragmentation. Thus, to achieve the high mass accuracy for the considered mass range (to assign exact mass and chemical formula), *m*/*z* = 21.022 (H_3_O^+^), *m*/*z* = 59.049 (C_2_H_5_O_2_^+^) and *m*/*z* = 137.132 (C_10_H_17_^+^) were used for internal calibration. Moreover, to guarantee high mass accuracy and a rapid identification of compounds, an off-line calibration was performed following the procedure described by [47]. Then, the acquired data (TofDaq software, Tofwerk AG, Thun, Switzerland) were processed following the procedure reported by Taiti and co-worker (2017) [46] and the VOC emissions expressed in ppbv (part per billion by volume) according to the formula described by Lindinger et al. [48]. Subsequently, data were filtered excluding all peaks ascribed to water chemistry or other interfering ions (e.g., oxygen or nitrogen monoxide) and all signals whose concentration was lower than 0.25 ppbv (when the average value was ranged from 0.25 to 0.50 ppbv has been reported in Table A1 as trace compound (Tr). Finally, the tentative identifications (TI) of compounds were based on models of fragmentation available in the literature and compared with published VOCs emitted from peach and nectarine fruits.

### 3.4. Sensory Measurements

*Panel test.* Nine trained (according to ISO 1993) and expert panelists selected inside of the School of Agriculture—University of Florence with familiarity to peach and nectarines, evaluated the sensory attributes of all peach samples included in this study. The same panelists evaluated the attributes of fruits of all cultivars. Sensory descriptors used for peaches and nectarines evaluation were visual appearance, pulp firmness, sweetness, crunchiness, fibrousness, aroma intensity, acidity, astringency, juiciness and overall judgment. The intensity of each attribute was evaluated based on a five-point scale: 1 = extremely poor; 3 = poor; 5 = acceptable (limit of marketability); 7 = good; 9 = excellent) and it was preceded by a visual assessment, both on whole fruits (five per cultivar) and on peeled pieces.

*Consumer acceptability.* Each consumer panel consisted around of 100 (50% women and 50% men aged between 21 and 70 years and selected) habitual (weakly) peach consumers. All consumers were recruited from staff and students at Florence University through personal communication and e-mail. Each tasting session (n = 50–60) was conducted over a span of two days. Consumer acceptance was measured as both degree of liking (1–9) and percentage of acceptance. When possible, each consumer assessed in different evaluation sessions all the peach variety samples and was asked to indicate his/her degree of liking/disliking using a 9-category hedonic scale (1—dislike extremely to 9—like extremely) [49]. A score of six was considered a commercial quality limit and the percentage of consumers liking was calculated as the percentage of consumers who liked the sample, with scores > 5 [50]. Finally, before to assess the consumer acceptability was done a visual evaluation, both on whole fruits (five for cultivars) and on peeled pieces.

### 3.5. Data Analysis

In order to observe the ordination of the samples (i.e., peach fruits suitable for short or long supply chains) in the chemical–physical variables space and in the panel test variables space, two different Principal Component Analyses (PCA) have been conducted. The PCA is an ordination technique based on a projection method that allows the display of information in a data matrix considering the influence of each in a limited number of components expressed as linear function of the original ones [51]. The first two (i.e., the most informative) principal components (PC) of each analysis have been reported. The results allow us to visualize the samples with their scores together with variables reported as vectors in a same ordination plot (biplot). The relations between samples and variables have been discussed. Another ordination technique, the Canonical Correspondence Analysis (CANOCO) was performed to observe the samples (i.e., peach fruits belonging to short or long supply chains) in same space considering two different kinds of datasets. We reported the ordination of the samples in *i.* the panel test and chemical-physical variables spaces and *ii.* the VOCs and panel test variables spaces. Canonical Correspondence Analysis [52] is a correspondence analysis of a matrix (VOCs in this case) where each sample has given values for one or more variables of a different kind (chemical–physical variables or panel test variables). The ordination axes are linear combinations of the environmental variables. CANOCO is thus an example of direct gradient analysis, where the gradient in chemical–physical variables or VOCs variables is known a priori and the panel test values are a response to this gradient. The final graphical report of the CANOCO is a triplot, which represents altogether the samples, the panel test and the third group of variables (chemical–physical or VOCs).

Analysis of variance (ANOVA) was performed to evaluate statistical significance differences between nectarines and peaches considering each of the morphological and physicochemical traits. ANOVA was performed with the software PAST (version 2.17v).

## 4. Conclusions

By the comparison among different peaches and nectarines varieties suitable for long supply chains and short supply chains, we have observed (1) important differences for physicochemical parameters, VOCs, sensory and consumer evaluation; (2) strong relationship among sensory evaluation, physicochemical traits and volatile compound profiles; and (3) deep differences between fruits suitable to LSCPs compared to SSCPs ones. The differences for aroma, sweetness and juiciness observed in “Regina di Londa” and “Guglielmina” compared to LSCPs ones, seemed to reflect the overall judgment expressed by the panelists. Moreover, by the consumer test it emerged how all varieties (harvested at physiological maturation) have satisfied the consumer tastes, albeit there were clear differences among the varieties studied and in particular between SSCPs and LSCPs. Once again, aroma and sugar content were the main parameters that drive the consumers to like the SSCPs more. Indeed, the CANOCO applied on VOCs data highlighted specific compounds were also typical of some cultivars, so they could be used drive the consumer’s preferences. On the contrary, the LSCPs varieties showed interesting values for firmness and crunchiness as required by the large retailers’ organization. Future studies need to better understand the different quality within the peaches suitable for the short chain, and to investigate which, among the newly varieties, could be suitable for the short chain.

## Figures and Tables

**Figure 1 foods-10-01253-f001:**
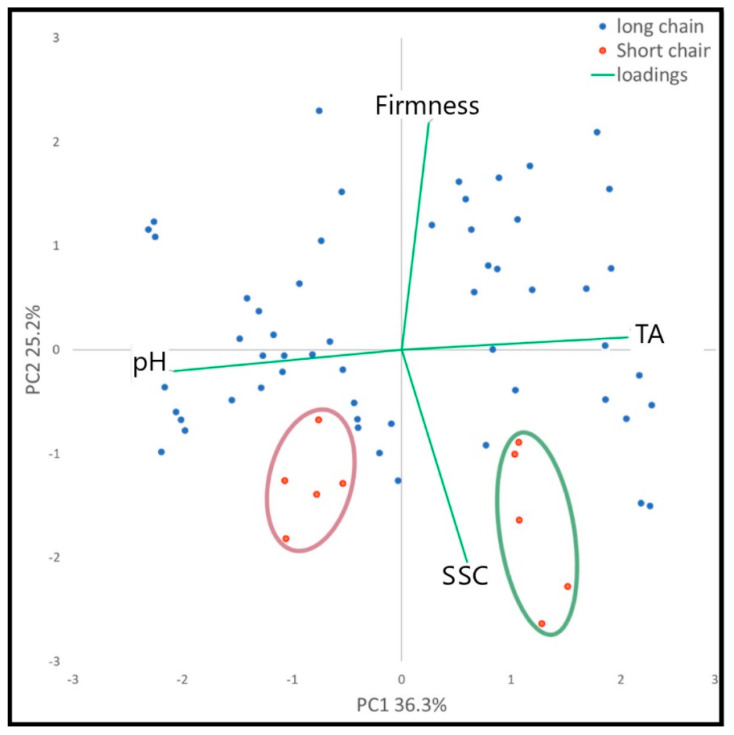
Scatter plot of the first two axes of the PCA conducted on physicochemical measures. Red circle highlights the “Guglielmina” samples; the green circle the “Regina di Londa” samples.

**Figure 2 foods-10-01253-f002:**
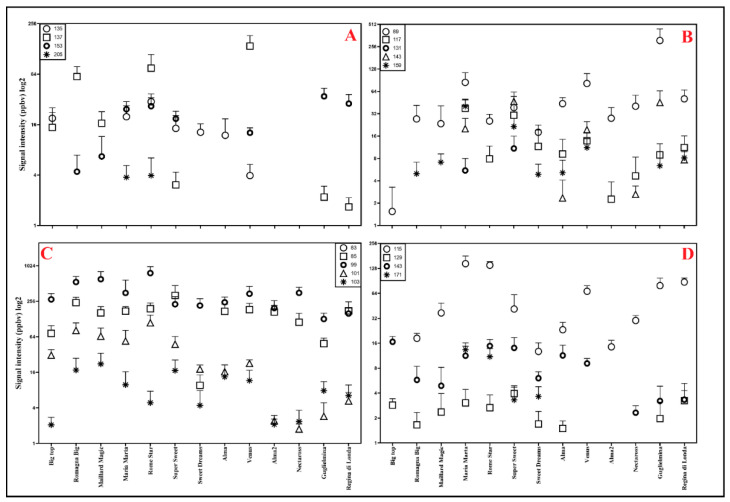
Signal intensities of different VOC types that possess strong peach-like aromas or are identified as impact volatiles in different varieties of peach such us: (**A**) Terpene compounds (*m*/*z* 135.117, 137.132, 153.127, 205.195); (**B**) Ester compounds (*m*/*z* 89.059, 117.091, 131.106, 143.107, 159.140), (**C**) C6 compounds (*m*/*z* 83.086, 85.101, 99.080, 101.096, 103.075), (**D**) Lactone compounds (*m*/*z* 115.075, 129.091, 143.107, 171.137) [34,35].

**Figure 3 foods-10-01253-f003:**
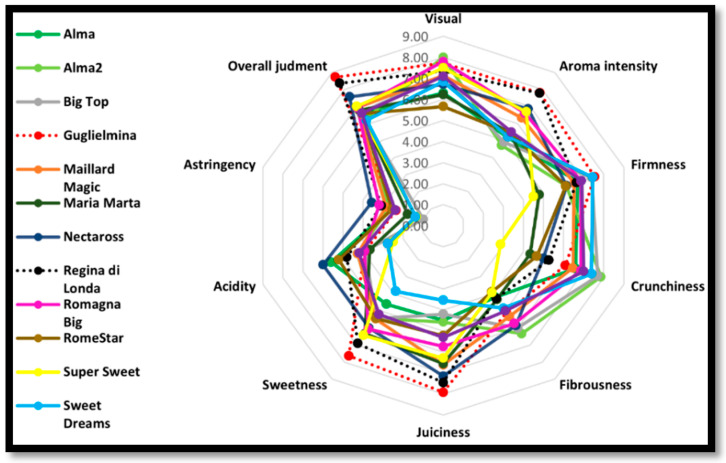
Schematic representation of the average values of the scores of ten attributes used for the sensory analysis.

**Figure 4 foods-10-01253-f004:**
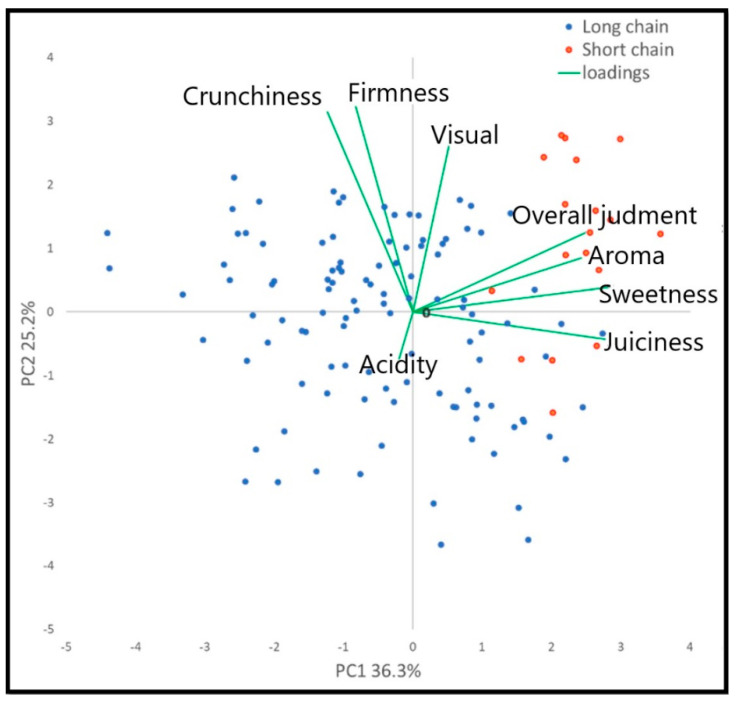
Scatter plot of the first two axes of the PCA conducted on panel test variables.

**Figure 5 foods-10-01253-f005:**
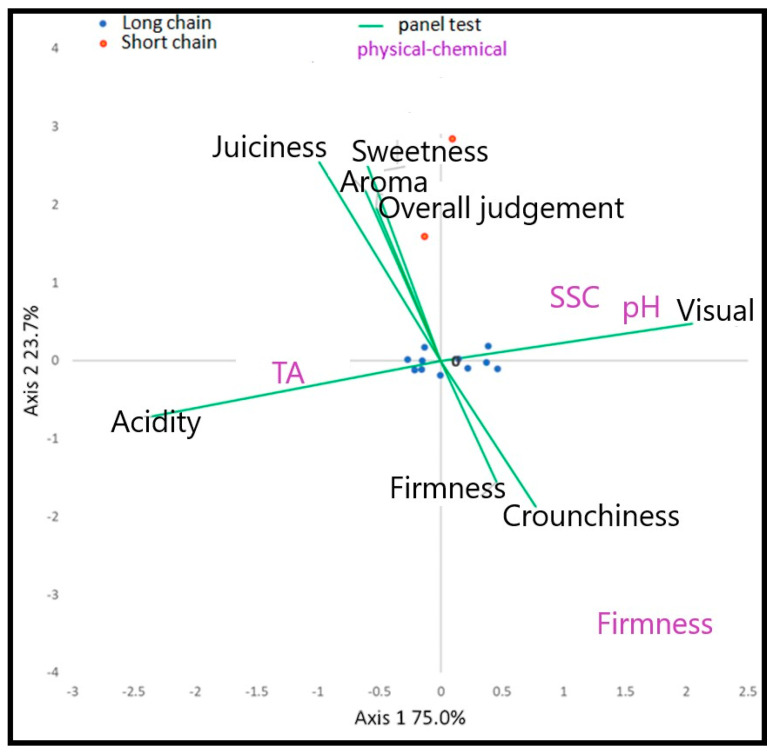
Triplot of the first two axes of the CANOCO conducted on physical–chemical (purple values) and panel test matrices (green vectors).

**Figure 6 foods-10-01253-f006:**
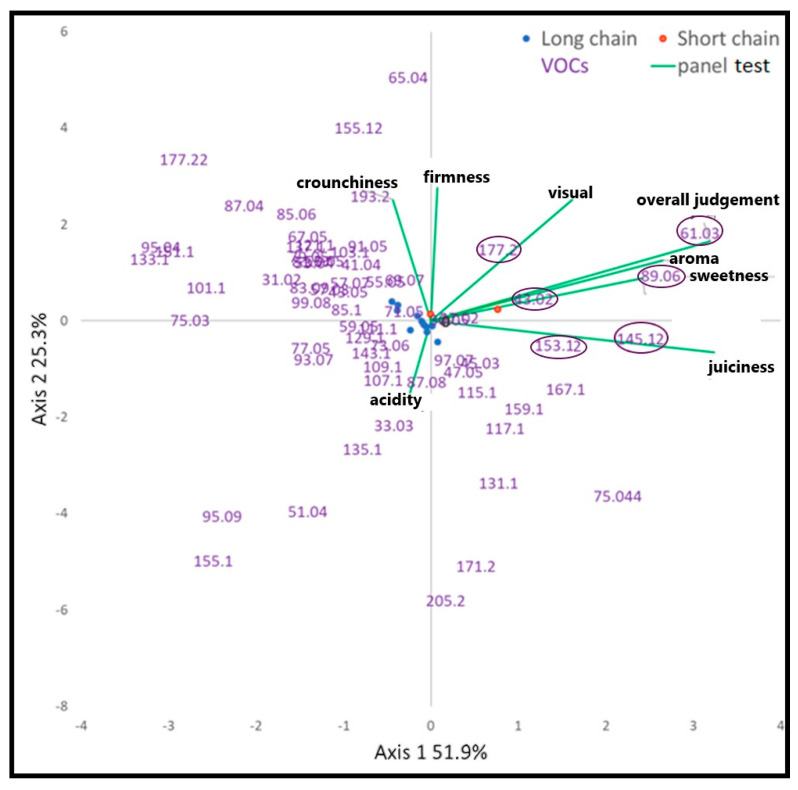
Triplot of the first two axes of the CANOCO conducted on panel test (purple values) and VOCs matrices (green vectors).

**Table 1 foods-10-01253-t001:** Morphological parameters: morphological traits (fruit average, Ø and height) and skin color (L, a, b). For morphological traits, different letters in the same row (nectarine and peaches grand total) denote significant differences among samples means (*p* < 0.05) (ANOVA).

Cultivar	Typology	Morphological Traits			Skin Color		
		Average Fruits Weight (g)	Ø Max (mm)	Height (mm)	L	a	b
Alma	Nectarine	295.5 (±24.8)	80.4 (±2.1)	76.3 (±5.3)	80.2 (±2.6)	−3.7 (±1.8)	52.3 (±3.5)
Alma2	Nectarine	254.1 (±28.3)	79.2 (±3.6)	75.0 (±3.7)	75.3 (±3.2)	−1.8 (±3.8)	54 (±2.3)
Big top	Nectarine	226.0 (±4.6)	91.7 (±13.7)	64.7 (±6.9)	47.5 (±10.9)	13.8 (±1.7)	31.9 (±9.3)
Guglielmina	Peach	242.0 (±15.2)	74.6 (±7.6)	67.2 (±5.5)	55.4 (±5.0)	20.7 (±2.5)	44.1 (±4.5)
Maillard Magic	Nectarine	209.2 (±26.7)	73.7 (±4.7)	68.8 (±3.1)	52.1 (±62.8)	24.3 (±8.2)	26.1 (±2.3)
Maria Marta	Peach	209.0 (±15.1)	74.7 (±2.5)	64.1 (±2.3)	57.1 (±5.5)	23.1 (±4.8)	41.9 (±4.9)
Nectaross	Nectarine	170.0 (±10.2)	69.0 (±2.4)	64.0 (±3.2)	77.5 (±3.5)	−2.4 (±2.9)	54.7 (±1.8)
Regina di Londa	Peach	256.0 (±16.4)	79.3 (±5.4)	72.0 (±2.2)	52.1 (±3.2)	23.3 (±3.4)	31.9 (±9.3)
Romagna Big	Nectarine	213.0 (±25.9)	73.4 (±3.4)	73.1 (±3.5)	65.0 (±5.4)	10.3 (±8.0)	46.0 (±3.9)
Rome Star	Peach	224.0 (±21.8)	76.0 (±3.4)	69.1 (±4.2)	40.2 (±7.9)	22.3 (±5.1)	20.2 (±5.8)
Lady Erica	Nectarine	312.0 (±15.1)	83.0 (±1.9)	80.6 (±4.8)	48.8 (±9.4)	27.7 (±5.2)	30.9 (±10.4)
Sweet Dream	Peach	277.5 (±26.6)	84.2 (±3.2)	73.0 (±2.2)	44.2 (±9.3)	18.8 (±2.3)	21.1 (±10.2)
Venus	Nectarine	260.0 (±27.9)	77.0 (±2.4)	76.3 (±5.8)	76.6 (±3.3)	−2.5 (±3.4)	55.4 (±1.6)
Nectarines average		250.5 (±48.4) ^a^	79.0 (±4.5) ^a^	73.0 (±3.7) ^a^			
Peaches average		244.5 (±48.4) ^a^	78.7 (±8.3) ^a^	70.5 (±6.1) ^a^			

**Table 2 foods-10-01253-t002:** Physicochemical parameters: pulp firmness (kgf), Soluble Solids Content (SSC, °Brix), pH, Titratable Acidity (meq/100 g fresh pulp), SSC-TA ratio. Different letters in the same row (nectarine and peaches grand total) denote significant differences among samples means (*p* < 0.01) (ANOVA).

Cultivar	Typology	Pulp Firmness (kgf)	SSC (°Brix)	pH	Titratable Acidity (meq/100 g Pulp FW)	SSC/TA Ratio
Alma	Nectarine	4.20 (±0.4)	13.3 (±0.9)	3.3 (±0.05)	42.3 (±0.3)	0.31
Alma2	Nectarine	4.10 (±0.6)	14.3 (±1.0)	4.5 (±0.1)	13.2 (±1.1)	1.08
Big top	Nectarine	4.30 (±0.5)	13.3 (±1.5)	3.1 (±0.4)	29.1 (±0.7)	0.46
Guglielmina	Peach	2.22 (±0.5)	14.6 (±1.5)	4.5 (±0.1)	17.4 (±0.5)	0.83
Maillard Magic	Nectarine	2.90 (±0.7)	12.5 (±0.8)	4.5 (±0.4)	17.7 (±2.4)	0.71
Maria Marta	Peach	1.6 (±0.5)	11.9 (±0.8)	4.2 (±0.1)	26.2 (±0.5)	0.45
Nectaross	Nectarine	3.4 (±0.3)	16.0 (±1.2)	3.5 (±0.1)	47.8 (±2.7)	0.34
Regina di Londa	Peach	1.5 (±0.5)	16.4 (±1.4)	3.5 (±0.1)	31.6 (±0.5)	0.52
Romagna Big	Nectarine	4.1 (±0.5)	15.2 (±1.2)	4.2 (±0.2)	25.4 (±2.5)	0.60
Rome Star	Peach	3.3 (±0.7)	13.1 (±0.6)	3.6 (±0.1)	32.8 (±0.7)	0.40
Lady Erica	Nectarine	2.5 (±0.4)	14.1 (±0.8)	4.9 (±0.1)	11.4 (±0.3)	1.24
Sweet Dream	Peach	3.9 (±0.6)	11.7 (±0.6)	5.2 (±0.3)	8.6 (±0.6)	1.36
Venus	Nectarine	4.0 (±0.5)	11.9 (±0.6)	3.5 (±0.05)	35.5 (±1.1)	0.33
Nectarines average		3.7 (±0.9) ^a^	13.5 (±1.5) ^a^	4.1 (±0.6) ^a^	25.6 (±11.9) ^a^	0.66
Peaches average		2.5 (±1.1) ^b^	14.0 (±1.9) ^a^	3.9 (±0.6) ^a^	27.0 (±8.4) ^a^	0.55

**Table 3 foods-10-01253-t003:** Acceptance of difference peach and nectarine fruits by Italian consumers harvested at “optimal maturity stage”. * Degree of liking: 1 = dislike extremely, 2 = dislike very much, 3 = dislike moderately, 4 = dislike slightly, 5 = neither like nor dislike, 6 = like slightly, 7 = like moderately, 8 = like very much, 9 = like extremely.

Variety	Degree of Liking * (1–9) (Average Value)	Acceptance (%)	Neither Like nor Dislike (%)	Dislike (%)
Alma	6.40	58%	24%	18%
Alma2	6.14	50%	28%	22%
Big Top	6.65	68%	32%	0%
Guglielmina	8.73	100%	0%	0%
Maillard Magic	6.93	71%	21%	8%
Maria Marta	7.00	59%	37%	4%
Nectaross	7.43	86%	14%	0%
Regina di Londa	8.36	100%	0%	0%
Romagna Big	6.94	74%	26%	0%
Rome Star	6.55	52%	46%	12%
Lady Erica	6.79	78%	22%	0%
Sweet Dream	6.08	50%	29%	21%
Venus	6.60	64%	28%	8%

**Table 4 foods-10-01253-t004:** General information: fruit type, fruit flesh color, fruit shape, diffusion area (^a^ Descriptor list for *Peach* (ECPGR Priority Descriptors for Peach [44])), ripening time (** “Big Top” fruits are collected in Faenza (Bologna, Emilia Romagna, Italy) at 5 July), Length of Supply Chain (LSC has been defined: Long ~25 days of shelf life under storage condition; Short ~3 days of shelf life under storage condition), and Perceived Acidity (PA has been defined as: S = Subacid (with acidity below 70 meq/L), B = Balanced (acidity between 70 and 130 meq/L), A = Acidic (acidity above 130 meq/L) [43].

Varieties	Fruit Type	Fruit Flesh	Fruit Shape ^a^ (Longitudinal Section)	Diffusion Area	Ripening Time (±Days of Collection Compared to “Big Top”) **	Perceived Acidity	Length of Supply Chain
Alma	Nectarine	Yellow	Round	Italy	+20	B	LSCP
Alma2	Nectarine	Yellow	Round	Italy	+32	S	LSCP
Big Top	Nectarine	Yellow	Round	International	/	S	LSCP
Guglielmina	Peach	Orange yellow	Round	Italy	+40	B	SSCP
Maillard Magic	Nectarine	White	Round	International	+5	S	LSCP
Maria Marta	Peach	Light yellow	Round	Italy	+15	B	LSCP
Nectaross	Nectarine	Yellow	Ovate	Italy	+20	A	LSCP
Regina di Londa	Peach	White	Round	Local—Tuscany	+50	B	SSCP
Romagna Big	Nectarine	Orange yellow	Round	Italy	+20	B	LSCP
Rome Star	Peach	Light yellow	Ovate	International	+20	B	LSCP
Lady Erica	Nectarine	Yellow	Round	Italy	+50	S	LSCP
Sweet Dream	Peach	Light yellow	Round	Italy	+40	S	LSCP
Venus	Nectarine	Light yellow	Ovate	Italy	+35	A	LSCP

## Data Availability

The datasets generated for this study are available on request to the corresponding author.

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
