# Peer review of "Are Peach Cultivars Used in Conventional Long Food Supply Chains Suitable for the High-Quality Short Markets?"

_foods, 2021, doi:10.3390/foods10061253_

Round 1

Reviewer 1 Report

Review of “Are peach cultivars used in conventional long food supply chains suitable for the high quality short markets?” by Taiti et al.

The paper examines 13 cultivars of peaches and nectarines for their suitability for immediate marketing after harvest.  The authors claim, with some truth, that stone fruit breeders would like to find cultivars that can be shipped to distant markets, and that this often impinges on the overall taste of the fruit.  They present a good study of the physico-chemical components compared to a panel of tasters, and statistics showing correlations between particular components and the taste rating. 

There are minor problems with the English, some of which I list below. 

Ln 64….highly…

Ln 66.  I would say, in contrast to rather than as

Ln 77… thus avoiding the need to manage…

Ln 86….Long Food Supply Chains (LFSCs), in the Short Food Supply Chains (SFSC)…

Ln 140. Remove the word with

Ln 142. Remove the word two

Ln 152.  Write out the full expression for VOC the first time you use this abbreviation.  This is done in Methods, but Methods comes at the end of the paper.

Ln 154…variety…

Ln 166….not shown…

Ln 167…among the cultivars there are difference in many VOCs types and intensities. are shown

ln 168…. As in other fruits,

ln 182.  Aroma is a ….

Ln 195.  Figure 2 reports the intensity…

Ln 213…combination among sweetness…

Ln 234…of the past 30-40 years have been…

Ln 241…in order to guarantee a long shelf-life…

Ln 244-5… when commercial maturity is reached, since the ripening process continues during post-harvest shipping and marketing.

Ln 250…shown…

Ln 279.  What is CANOCO?  This is explained in the Methods, but you encounter it first in Results and it should be explained here as well.

Ln 319….wide range….

Page 12.  Perceived Acidity (PA has been defined as: S=Subacid (with acidity below 70 meq/l), B=Balanced (acidity between 70 and 130 meq/l), A=Acidic (acidity above 130 meq/l) [22].

The Figure legends are under the wrong figures which makes it confusing.  Figure 3 is written out in full in the text while the other figures are abbreviated to Fig.  

Author Response

The paper examines 13 cultivars of peaches and nectarines for their suitability for immediate marketing after harvest.  The authors claim, with some truth, that stone fruit breeders would like to find cultivars that can be shipped to distant markets, and that this often impinges on the overall taste of the fruit.  They present a good study of the physico-chemical components compared to a panel of tasters, and statistics showing correlations between particular components and the taste rating. 

REPLY: We have very appreciated the referee comments.

There are minor problems with the English, some of which I list below. 

REPLY: We have now correct all the “English problems” as suggested, and we have revised the text.

Ln 279.  What is CANOCO?  This is explained in the Methods, but you encounter it first in Results and it should be explained here as well.

REPLY: Done

Page 12.  Perceived Acidity (PA has been defined as: S=Subacid (with acidity below 70 meq/l), B=Balanced (acidity between 70 and 130 meq/l), A=Acidic (acidity above 130 meq/l) [22].

REPLY: Done

The Figure legends are under the wrong figures which makes it confusing.  Figure 3 is written out in full in the text while the other figures are abbreviated to Fig.  

REPLY: The term "Figure" and “Table” has been used in full in the text.

Reviewer 2 Report

Are peach cultivars used in conventional long food supply chains suitable for the high quality short markets?” is very interesting and well written.

The manuscript authors the introduction section presented the current state of knowledge on the experimental design. The topic is a very new, because touch a problem of peaches and nectarines  composition and quality.

The Introduction section includes all necessary information about examined objects and problems.

Results section are presented in clear and easily way.

One question to the Authors:

Table 1: in my opinion headline description should be put above table not below. On the other hand this table is too big (too much information is inside) so it is a very difficult for reading. Please could you divided it for two smaller tables, as well paste it in described section. Not at the end of manuscript.

Why in presented table there is no statistical tools present (p-value and letters for homogeneous groups), please correct it.

Similar remark to all figures, paste it to adequate part of description manuscript part.

The discussion section presents a very good comparison of the obtained results with other results available in the data basis.

General opinion: I think, that presented manuscript is a very valuable with average scientific value.

Author Response

The manuscript authors the introduction section presented the current state of knowledge on the experimental design. The topic is a very new, because touch a problem of peaches and nectarines  composition and quality.

The Introduction section includes all necessary information about examined objects and problems.

Results section are presented in clear and easily way.

REPLY: Thank you for your comments.

One question to the Authors:

Table 1: in my opinion headline description should be put above table not below. On the other hand this table is too big (too much information is inside) so it is a very difficult for reading. Please could you divided it for two smaller tables, as well paste it in described section. Not at the end of manuscript.

REPLY: We agree with the referee and we changed as suggested. Thus, Table 1 has been split into two smaller tables. Moreover, all Table and Figure were correctly moved in the text.

Table 1: Morphological parameters: morphological traits (fruit average, Ø and height) and skin color (L, a, b). For morphological traits, different letters in the same row (nectarine and peaches grandtotal) denote significant differences among samples means (p < 0.05) (ANOVA).

Table 2 - Physico-chemical parameters: pulp firmness (kgf), Soluble Solids Content (SSC, °Brix), pH, Titratable Acidity (meq/100 g fresh pulp), SSC-TA ratio. Different letters in the same row (nectarine and peaches grandtotal) denote significant differences among samples means (p < 0.01) (ANOVA).

Why in presented table there is no statistical tools present (p-value and

letters for homogeneous groups), please correct it.

REPLY: We agree with the referee and we changed as suggested.

Peach and nectarine statistical differences has been highlighted with letters (Table 1 and 2). In section 3.6. (Data analysis) it has been added: Analysis of variance (ANOVA) was performed to evaluate statistical significance differences between nectarines and peaches considering each of the morphological and physico-chemical traits. ANOVA was performed with the software PAST (version 2.17v). Moreover, The following sentences have been added in the text Line 105-107: Between peach and nectarines grand total no significant differences were highlighted in morphological traits.

Similar remark to all figures, paste it to adequate part of description manuscript part.

REPLY: We agree with the reviewer and we changed as suggested

The discussion section presents a very good comparison of the obtained results with other results available in the data basis.

General opinion: I think, that presented manuscript is a very valuable with average scientific value.

REPLY: we want to thank the referee for his comments.

Round 2

Reviewer 2 Report

Are peach cultivars used in conventional long food supply chains suitable for the high quality short markets?” is very interesting and well written.

The manuscript authors the introduction section presented the current state of knowledge on the experimental design. The topic is a very new, because touch a problem of peaches and nectarines composition and quality.

Authors corrected manuscript according to my points and remarks. I am a satisfied.

General opinion: I think now, after correction, manuscript is a very valuable with good and positive scientific value. Manuscript is good to publish in present form in Foods journal.

This manuscript is a resubmission of an earlier submission. The following is a list of the peer review reports and author responses from that submission.